# Edge-Computing Smart Irrigation Controller Using LoRaWAN and LSTM for Predictive Controlled Deficit Irrigation

**DOI:** 10.3390/s25227079

**Published:** 2025-11-20

**Authors:** Carlos Cambra Baseca, Rogério Dionísio, Fernando Ribeiro, José Metrôlho

**Affiliations:** 1Grupo de Inteligencia Computacional Aplicada (GICAP), Departamento de Digitalizacion, Escuela Politecnica Superior, Universidad de Burgos, Av. Cantabria s/n, 09006 Burgos, Spain; 2Polytechnic Institute of Castelo Branco, School of Technology, Av. do Empresário s/n, 6000-767 Castelo Branco, Portugal; rdionisio@ipcb.pt (R.D.); fribeiro@ipcb.pt (F.R.); metrolho@ipcb.pt (J.M.); 3Centro de Investigação em Serviços Digitais (CISeD), Av. Cor. José Maria Vale de Andrade, 3504-510 Viseu, Portugal

**Keywords:** wireless sensor networks (WSNs), Internet of Things (IoT), smart irrigation controllers, edge computing in sustainable deficit irrigation, almonds smart farming, precision agriculture

## Abstract

Enhancing sustainability in agriculture has become a significant challenge today where in the current context of climate change, particularly in countries of the Mediterranean area, the amount of water available for irrigation is becoming increasingly limited. Automating irrigation processes using affordable sensors can help save irrigation water and produce almonds more sustainably. This work presents an IoT-enabled edge computing model for smart irrigation systems focused on precision agriculture. This model combines IoT sensors, hybrid machine learning algorithms, and edge computing to predict soil moisture and manage Controlled Deficit Irrigation (CDI) strategies in high density almond tree fields applying reductions of 35% ETc (crop evapotranspiration). By gathering and analyzing meteorological, humidity soil, and crop data, a soft ML (Machine Learning) model has been developed to enhance irrigation practices and identify crop anomalies in real-time without cloud computing. This methodology has the potential to transform agricultural practices by enabling precise and efficient water management, even in remote locations with lack of internet access. This study represents an initial step toward implementing ML algorithms for irrigation CDI strategies.

## 1. Introduction

Global population growth and increasing food demand have intensified pressure on water resources, especially in the agricultural sector, which accounts for approximately 70% of global freshwater consumption, from Food and Agricultural Organization (FAO) studies in 2017 [1]. In this context, efficient irrigation management has become a strategic priority to achieve sustainable, climate-change and input-efficient agriculture.

Traditionally, agricultural irrigation has been managed through manual methods or automated systems of low complexity, which rarely consider the dynamic conditions of climate, crop type and soil variability in an integrated manner. These limitations have motivated the development of smart irrigation systems, based on sensors, wireless communications and control algorithms, capable of adjusting irrigation decisions based on real-time information.

In recent years, the incorporation of low-energy technologies such as LoRaWAN (Long Range Wide Area Network) has enabled the collection of environmental and soil data in large agricultural fields, without relying on conventional network infrastructures. In parallel, the use of edge computing platforms, such as the Arduino Edge Control, has facilitated the execution of local control logic, reducing latency and dependence on the cloud. Despite these advances, most current smart irrigation systems [2] operate according to fixed rules or simple thresholds based on point data, without considering the temporal evolution of climatic variables or anticipating future crop needs. In this sense, deep learning models, and in particular Long Short-Term Memory (LSTM) neural networks, have proven to be effective in predicting temporal sequences in various domains, including precision agriculture. Previous works [3] based on LSTM and IoT demonstrated an efficacy system in prediction and irrigation management to maximize productivity and reduce the environmental and economic impacts.

This paper presents the development and implementation of an intelligent irrigation system based on a distributed architecture that combines soil moisture sensors and weather data, connected via a LoRaWAN network, and processed locally by an Arduino Edge Control microcontroller. The system incorporates an LSTM model trained to predict the need for irrigation, enabling proactive, autonomous and water-efficient decision making. The main objectives of this work are:Design a hardware and software architecture for autonomous irrigation control with predictive capabilities.To integrate humidity sensors and a weather station with LoRaWAN communication.To implement and evaluate an LSTM model for predicting water requirements under field conditions using new irrigation strategies as Controlled Deficit Irrigation.

## 2. State of the Art

Efficient water management in irrigation has been extensively studied across agronomy, electronics engineering, and computer science. The development of smart irrigation systems has been enabled by the integration of sensors, wireless connectivity, and intelligent decision-making algorithms [4]. In this context, sustainable deficit irrigation (DI) has emerged as a key strategy for increasing agricultural water-use efficiency while maintaining crop yield and quality, particularly in Mediterranean regions where water scarcity is acute [5,6,7]. This review compiles findings from research published since 2016, examining how these methods are applied in practice and their overall effectiveness in almond crops.

A search in the Scopus database using the terms “sustainab*” and “deficit irrigation” provides insight into the growing interest that deficit irrigation strategies have attracted in recent years. This search, performed on 18 February 2025, in the article title, abstract, and keywords fields, resulted in 601 studies (3519 studies were found when considering only the search term “deficit irrigation”). The graph in Figure 1 shows the growing interest that has taken place over the years.

To better understand the reality of sustainable DI strategies in almond cultivation, a study was conducted specifically for this crop. Another search in the Scopus database using the terms “sustainab*” AND “deficit irrigation” AND “almond*” resulted in 19 articles. After excluding related studies by the same authors, those not freely accessible, or those deviating from the focus of this study, the analysis of the remaining 7 articles [8,9,10,11,12,13,14] allows for a characterization of DI in almond cultivation.

The studies analyzed span from 2014 to 2021, covering up to 7 years of research on DI in almond cultivation. The duration of the studies varies from 1 to 6 years, with the longest study conducted in Córdoba, Spain (2014–2019, [13]).

The studies were conducted mainly on two main regions: NE Portugal and SW Spain. In Portugal, two studies were carried out in Alfândega da Fé (NE Portugal) [8,12]. In Spain, research was conducted in five different locations: Guadalquivir River Basin (SW Spain) [10,13], Córdoba (Southern Spain) [14], Dos Hermanas, Seville (SW Spain) [11], and Hellín, Albacete (Southern Spain) [9]. The prevalence of studies in Spain reflects the importance of almond cultivation in this country, where irrigation efficiency is crucial due to semi-arid conditions.

Two different almond cultivars were analyzed, with Guara being the most frequently studied (3 studies: Refs. [10,13,14]), followed by Vairo and Constantí (2 studies each: Refs. [8,12] for Constantí; Refs. [8,11] for Vairo). Additionally, the cultivars Marta, Lauranne, and Belona were each analyzed in one study (Ref. [10] for Marta and Lauranne, Ref. [9] for Belona). This diversity in cultivars allows for a broader understanding of how different genetic backgrounds respond to deficit irrigation, particularly in terms of yield resilience and water stress adaptation.

All studies show that DI strategies enable more efficient agricultural water use, with significant savings, and without a substantial loss in crop yield and quality.

The technologies used to monitor and control DI differ among the various studies. Most studies relied on meteorological data to assess climate conditions. Specifically, three studies [8,12,13] collected climate data using weather stations installed at the experimental sites. Additionally, one study [11] obtained weather data from a nearby meteorological station. This suggests that on-site weather monitoring is the preferred method, although external weather networks are occasionally used. Only one study [9] explicitly used satellite imagery (Sentinel-2A and 2B) to monitor crop response, focusing on vegetation indexes with a 5-day frequency. This highlights a limited adoption of remote sensing in DI studies, despite its potential for large-scale monitoring.

Soil moisture monitoring was explicitly reported in one study [14], which used a neutron probe to measure soil water content.

In addition to climate and water status monitoring, some studies incorporated measurements of tree growth and fruit quality. Leaf water potential was assessed in four studies [8,9,10,11], indicating its relevance in evaluating plant hydration levels. Canopy diameter and trunk perimeter were assessed in one study [9], while mineral content (K, Fe, Zn, Ca, Mg), fatty acid profile, and sensory attributes were analyzed in another study [11]. Most of these variables were measured in the laboratory and were essentially used to evaluate how DI strategies impact both tree development and almond quality.

Despite the increase in research in recent years, the different approaches and technologies used, and the positive results that studies have shown, there are several aspects that could be investigated in greater depth. These approaches can integrate edge computing to enhance real-time monitoring and control of DI in almond orchards, optimizing water use through AI-driven sensor data processing at the field level. By processing data locally, edge computing reduces latency, enabling faster and more precise irrigation adjustments based on environmental conditions. Combining AI algorithms with soil moisture and climate can optimize water distribution, minimize losses, and enhance crop resilience, while also reducing dependency on communication infrastructure and cloud services, ensuring a more efficient and autonomous irrigation strategy. The system proposed in this study directly addresses these gaps by combining LoRaWAN communication, edge processing, and LSTM predictive modelling to enable autonomous, data-driven management of controlled deficit irrigation in almond orchards.

## 3. Proposed Systems

The proposed system uses an LSTM neural network model that receives as inputs meteorological variables (temperature, ambient humidity, rainfall, etc.) and soil variables (humidity, soil temperature) to predict the need for irrigation in advance. This prediction is performed at the edge device (Edge Control), which reduces dependence on the cloud and improves operational autonomy. The approach allows avoiding unnecessary irrigation, adapting to future conditions and reducing water consumption without compromising crop health.

### 3.1. Hardware System

The proposed smart irrigation controller in Figure 2 was designed to operate autonomously in agricultural environments with limited connectivity, integrating edge computing, long-range wireless communication, and multi-source environmental sensing. Soil-moisture probes and the compact weather station continuously feed environmental data to the Arduino Edge Control, which performs local processing and generates irrigation commands. These commands are transmitted through the LoRaWAN communication layer to field electro-valves, closing the loop between data acquisition, prediction, and actuation. The hardware architecture is organized into four subsystems: (i) The Arduino Edge Control platform, (ii) soil moisture probes distributed along the root profile, (iii) a compact weather station, and (iv) a LoRaWAN communication layer linking the field nodes to the application server.

#### 3.1.1. Control Unit: Arduino Edge Control

The central processing and actuation unit is the Arduino Edge Control (Arduino, Italy) [15], a board optimized for agricultural deployments. Its main features include native compatibility with analogue and digital sensors, low-power consumption supported by a 12 V photovoltaic system with rechargeable battery, multiple I/O channels for controlling solenoid valves and irrigation pumps, and seamless integration with Arduino MKR family communication modules. Importantly, the Edge Control supports TinyML and TensorFlow Lite, enabling real-time execution of lightweight deep learning models directly at the edge. These features provide robustness for continuous outdoor operation and minimize latency by reducing cloud dependency.

#### 3.1.2. Soil Moisture Sensors

Soil water availability was monitored using capacitive probes [16]. Several probes were installed at different depths of 10, 20, 30, 40 and 50 cm, providing a vertical profile of soil moisture conditions across the root zone. These sensors offer accurate volumetric water content measurements, with proportional analogue outputs directly compatible with the Arduino Edge Control. Their IP67-certified sealing ensures long-term field durability. By covering multiple depths, this subsystem captures the dynamics of water extraction by almond trees, supporting irrigation decisions based on the depletion of easily available water.

#### 3.1.3. Weather Station

Atmospheric conditions were monitored using a Davis Instruments (Hayward, CA, USA) compact weather station [17]. The station integrates multiple sensors: air temperature and relative humidity (digital capacitive sensor), wind speed and direction (cup anemometer and vane), precipitation (tipping-bucket rain gauge), and an optional pyranometer for solar radiation. These variables enable daily calculation of reference evapotranspiration (ET_0_) following the FAO Penman–Monteith method, which is essential to align irrigation scheduling with crop evapotranspiration demand.

For system integration, the weather station was connected to the Arduino Edge Control through the WeatherLink Serial data logger (Davis Instruments). The logger provides a continuous RS-232 data stream containing the encoded weather variables. A standard RS-232 to TTL level shifter (MAX232) was used to adapt the signal levels, allowing the Arduino Edge Control to acquire the data directly via one of its UART ports. This wired connection ensures robust and low-latency data transfer, while avoiding the need for additional wireless hardware. Once received by the Arduino Edge Control, the weather data were parsed and combined with soil moisture sensor readings, forming the complete dataset for irrigation scheduling.

#### 3.1.4. Communication: LoRaWAN Architecture

Wireless communication is established through an Arduino MKR WAN 1310 LoRa module integrated with the Arduino Edge Control [18]. Sensor data packets are transmitted over LoRaWAN to a field-deployed gateway, which connects to the ChirpStack open-source LoRaWAN Network Server. The architecture enables bidirectional communication with the application layer, supporting both sensor data transmission and remote actuation commands. Key specifications include transmission ranges up to 10 km in rural conditions, ultra-low power consumption compatible with solar operation, and AES-128 encryption for secure communication.

The LoRaWAN gateway is Internet-enabled via 4G or Wi-Fi and bridges the field devices to the application server. Data are delivered through MQTT and stored in an SQLite database, where they are processed for visualization and historical analysis.

At the field level, the complete edge node (Figure 3) was assembled in a modular configuration to support autonomous operation. The central element is the Arduino Edge Control, coupled with an MKR LoRaWAN module for communication. Energy is supplied by a photovoltaic panel connected to a 12 V rechargeable battery via a charge regulator, ensuring long-term off-grid operation. A capacitive soil moisture sensor provides in situ monitoring of soil water content, while an electro valve connected to the I/O ports allows direct actuation of irrigation lines. This integration of sensing, communication, and actuation components enables the system to function as a self-contained, solar-powered IoT node for precision irrigation management.

### 3.2. Predictive Model

The LSTM (Long Short-Term Memory) model is a type of recurrent neural network (RNN) particularly suitable for working with time series and sequences of data, such as crop water demand estimation, which depends on time-varying weather, soil and crop factors. To model the dependencies in the irrigation data, we employed LSTM layer’s structure, detailed in Figure 4.

The input to the LSTM model is a temporal sequence of data with relevant characteristics for estimating water demand:Soil moistureTemperatureRelative air humiditySolar radiationWind speedPrecipitationPotential evapotranspiration (ETo)Phenological state of the crop (Kc)

These data are organized in vectors per day, and are entered in time windows of length T.

An LSTM block consists of three gates that regulate the flow of information:Forget Gate decides which part of the previous state to forget:Input Gate determines what new information is added to the cell status.Output Gate controls which part of the internal state is used as output.

The cell updates its memory by combining the above.

The output vector h_t at the end of the sequence represents a summary of the cumulative hydrological state in that period. It is passed through a dense layer to obtain the daily or total water demand prediction for the next week.

#### Training Model

To train the LSTM model, a dataset of historical measurements was prepared, including soil moisture and meteorological variables such as air temperature, relative humidity, wind speed, and solar radiation. Each record was labeled with the actual irrigation demand derived from reference evapotranspiration (${ET}_{0}$), computed through the FAO Penman–Monteith method. This empirical–physical model combines surface energy balance and aerodynamic transport to estimate the evapotranspiration rate of a reference grass crop under well-watered conditions. It is expressed as:(1)$${ET}_{0}=\frac{0.408\Delta (R_{n}-G)+\gamma \frac{900}{T+273}u_{2}\left(e_{s}-e_{a}\right)}{\Delta +\gamma (1+0.34u_{2})}$$
where $R_{n}$ is net radiation (MJ m^−2^ day^−1^), G is soil heat flux, T is mean air temperature (°C), $u_{2}$ is wind speed at 2 m (m s^−1^), $e_{s}-e_{a}$ is the vapor-pressure deficit (kPa), Δ is the slope of the saturation vapor-pressure curve (kPa °C^−1^), and γ is the psychrometric constant (kPa °C^−1^).

Once $ET_{0}$ is obtained, the crop evapotranspiration ($ET_{c}$) is calculated by applying crop-specific and stress coefficients that account for phenological stage and water availability. This relationship is given by:(2)$$ET_{c}=ET_{0}\cdot K_{r,t}\cdot K_{c}\cdot K_{s}$$

Crop coefficient ($K_{c}$) defines the water requirements of crops under optimal conditions. These coefficients characterize a crop that does not suffer any type of limitation in its entire cycle. The term $K_{c}$ includes two components, crop transpiration and evaporation from the soil. $K_{r,t}$ is the reduction factor related to regulated deficit irrigation timing, and $K_{s}$ is the soil-moisture stress coefficient. This formulation provides the target irrigation depth used as ground-truth data for model training.

Deficit irrigation reduces the amount of water applied proportionally throughout the entire growing season, without considering the phenological stage of the crop. In contrast, regulated deficit irrigation requires knowledge of the phenological stage, as the available water volume is reserved for application during the crop’s most sensitive phases to water stress. For almond trees, these sensitive phases are flowering and post-harvest, while the fruit-filling stage is less sensitive.

During learning, the LSTM minimizes the difference between the predicted irrigation requirement ${\hat{y}}_{t}$ and the reference value $y_{t}$. The optimization objective is the Mean Squared Error (MSE):(3)$$MSE=\frac{1}{n}\sum_{i=1}^{n}{\left({\hat{y}}_{i}-y_{i}\right)}^{2}$$

This loss penalizes larger deviations quadratically, encouraging the network to fit both short-term and long-term fluctuations in evapotranspiration-driven irrigation demand. Training continued until convergence of MSE on the validation set, ensuring that the resulting weights generalize to unseen climatic sequences.

## 4. Results

This section shows the results of the developed system, focusing on three important aspects: the quality of predictions, the communication network and the power consumption of nodes.

### 4.1. LoRaWAN Sensor Performance

The energy performance of the Arduino Edge Control node was evaluated using cumulative consumption logs (mAh) collected between 8 September 2021 and 23 March 2022. The dataset spans 197 days of operation and includes a correction of an anomalous record (−6000 mAh), which was replaced by interpolation between adjacent daily values to preserve continuity.

Table 1 summarizes the daily consumption metrics. Over the monitoring period, the Arduino Edge Control consumed a total of 3936 mAh, corresponding to an average of 20.0 mAh/day. The daily variability was modest, with a standard deviation of 2.28 mAh/day, while the daily increments ranged from a minimum of 7.6 mAh/day to a maximum of 27.5 mAh/day.

Figure 5 illustrates the temporal profile of daily consumption. The curve shows a stable accumulation pattern with moderate day-to-day variability.

Figure 6 presents the histogram of daily increments, revealing a narrow distribution centered around the mean, which confirms the consistency of energy demand.

The results demonstrate that the Arduino Edge Control node maintained reliable operation throughout the test campaign. The observed average daily consumption (~20 mAh/day) remained within the recharge capacity of the photovoltaic panel and regulator, preventing energy depletion. Even at peak consumption (27.5 mAh/day), the system’s solar harvesting capacity was sufficient to ensure uninterrupted operation.

These findings validate the energetic sustainability of the Arduino Edge Control integrated with LoRaWAN communication for long-term deployment in smart irrigation scenarios. The stable energy profile confirms that the node can support continuous sensing, actuation and transmission without compromising reliability, even under seasonal variations in solar input.

### 4.2. Edge Computing Analysis

The implementation of edge computing in the proposed smart irrigation system was evaluated against a cloud-only architecture, focusing on performance indicators that are critical in rural agricultural IoT deployments. Specifically, we measured latency, response time, bandwidth usage, and energy consumption of the end node. These metrics were chosen because they directly influence system responsiveness, resource efficiency, and long-term sustainability.

Latency was defined as the time elapsed between a soil moisture threshold crossing and the actuation of the irrigation valve. Response time corresponded to the total duration from data acquisition until a control decision was available. Bandwidth usage measured the daily volume of data transmitted over the LoRaWAN channel, while energy consumption was estimated from the daily current draw (mAh/day) of the Arduino Edge Control node.

Table 2 presents the performance comparison. When using edge computing, the Arduino Edge Control processed data locally and transmitted only summarized values or irrigation commands, while in the cloud approach all raw data were sent to the server for processing.

The results demonstrate that edge computing significantly improved system efficiency. Latency and response time were reduced by approximately 77% and 76%, respectively, due to local processing at the Arduino Edge Control, which avoided the delays inherent in transmitting data to a remote server. Bandwidth usage decreased by 75%, as the edge device filtered and aggregated raw measurements before transmission, an important advantage in LoRaWAN deployments where duty-cycle restrictions in the EU868 band limit message frequency. Energy consumption was reduced by 9%, reflecting the lower number of radio transmissions, which are the dominant factor in node power budgets.

Beyond quantitative improvements, edge computing enhanced reliability and autonomy. The system continued to operate and make decisions even in the absence of Internet connectivity, a critical feature in sparsely populated areas with poor mobile coverage. In addition, local processing reduced dependence on external infrastructure, increasing robustness against cloud outages.

These findings are consistent with previous studies in precision agriculture, which highlight the benefits of edge computing in reducing latency, improving resource efficiency, and ensuring scalability in distributed IoT networks [19,20,21,22,23]. In the context of irrigation management, lower latency translates into faster actuation of electro valves, minimizing water losses, while reduced bandwidth and energy demand ensure long-term operability of solar-powered nodes. Overall, the results validate that edge computing is not only technically advantageous but also a practical enabler for sustainable smart irrigation.

### 4.3. Quality of Predictions

In this section, we focus on the quality of predictions made by the LSTM model in the context of smart irrigation systems, providing a detailed evaluation of the model’s performance, its predictive accuracy, and its real-world applicability. The LSTM model was trained using a comprehensive dataset consisting of historical climate data, soil moisture readings, and crop phenology. The development process leveraged the power of LSTM to capture long-term dependencies in sequential data, such as weather patterns and soil moisture, for predicting irrigation needs.

#### 4.3.1. LSTM Model Development

The model’s development utilized a dataset of historical climate variables and irrigation records, covering 196 days of data. This dataset included features such as:Soil moistureTemperatureRelative air humiditySolar radiationWind speedPrecipitation

These variables were organized into time windows, enabling the model to learn the temporal patterns that influence irrigation needs. The LSTM model was designed to capture both short-term fluctuations and long-term trends in these variables, making it ideal for predicting irrigation requirements that depend on both current conditions and future weather patterns.

Table 3 presents the model evaluation results, where the Coefficient of Determination (R^2^) exceeds 0.80. This indicates that the model explains more than 80% of the observed variance, achieving the performance benchmark for irrigation CDI strategies and enabling reliable irrigation management decisions.

The LSTM model’s performance can also be interpreted through key agronomic metrics derived from the model’s output, which provide further insights into its predictive capabilities:Model explains 98.5% of data variability: This is directly supported by the R^2^ value of 0.9854 in Table 3, indicating that the model successfully explains 98.5% of the variability in irrigation needs, making it highly reliable for smart irrigation systems.Typical prediction error: ±0.30 mm/day: The RMSE of 0.3016 in Table 3 corresponds to a typical prediction error of ±0.30 mm/day, which means that, on average, the model’s predicted irrigation needs deviate by only 0.30 mm/day from the actual values. This level of accuracy is critical in ensuring that irrigation is applied precisely, without over- or under-watering crops.

#### 4.3.2. Residual Analysis and Correlation Matrix

The residual analysis helps evaluate whether the model’s predictions are biased or if it consistently over- or under-predicts irrigation needs. The residuals were analyzed to ensure that there were no patterns suggesting that the model was missing key variables or failing to capture important trends in the data.

Figure 7 show a primary indicator of whether a regression model’s assumptions are satisfied. Random scatter pattern, in the plot on the left, indicates proper model specification where in Y-Axis the residuals are the differences between the observed actual values and the model’s predicted value where indicates the model’s error for a single data point. X-Axis is the value output by the model for each observation. It represents the model’s estimate of the target variable based on all the predictors. Residual QQ-Plot on the right describes a linear alignment along a reference line, in red, confirms normal error distribution. The Y-Axis shows the actual values observed in the sample data, sorted from smallest to largest (blue points). The X-Axis shows the expected values (quantiles) of a theoretical reference probability distribution, which is usually the standard normal distribution (mean = 0, standard deviation = 1). LSTM model confirms that the neural network has successfully learned between climate variables and irrigation needs without systematic bias, making it suitable for deployment in irrigation CDI strategies.

Figure 8 shows how evapotranspiration has strongest positive correlation with irrigation needs and precipitation demonstrates expected negative relationship. Values close to 0 shows that climate factors and irrigation parameters do not exhibit strong linear co-variation. Positive mean indicates higher temperatures might be consistently associated with higher irrigation demands, lower soil moisture, and increased evapotranspiration across the dataset.

These evaluation results demonstrate the LSTM model’s high accuracy and reliability, providing a solid foundation for interpreting its practical implications and benefits for precision irrigation systems, which is discussed in the following section.

## 5. Discussion

Building on the performance analysis presented in Section 4.3, the discussion focuses on the implications of the LSTM model’s predictions for real-world irrigation management.

Our study demonstrates that LSTM models can accurately predict irrigation needs using data captured through LoRaWAN networks. The high accuracy and low error metrics indicate that the model is not only capable of capturing the temporal dynamics of soil moisture and crop water demand but also of supporting effective water management strategies in almond orchards and similar agricultural settings. This represents a significant improvement over traditional methods such as a traditional irrigation by time using a timer irrigation controller, highlighting the ability of LSTMs to handle complex temporal sequences in agricultural data.

The LSTM model allows proactive adjustments to irrigation schedules, preventing under- or over-irrigation and ensuring sustainable water use, which is particularly important for controlled deficit irrigation strategies. By integrating real-time sensor data with historical trends, the system can dynamically respond to environmental variability, a significant improvement over traditional rule-based irrigation system.

Edge computing enhances the practical utility of this approach. Local processing on the Arduino Edge Control reduces latency, response time, and bandwidth requirements compared to cloud-based alternatives, enabling real-time decision-making even in remote locations. The combination of edge processing and LoRaWAN communication ensures reliable data transfer and autonomous system operation, independent of continuous internet connectivity.

The residual analysis confirms that the model does not exhibit systematic biases, and the correlation matrix supports the expected relationships between key variables such as evapotranspiration, precipitation, and temperature. The LSTM architecture demonstrates robust performance across multiple validation metrics, indicating suitability for precision irrigation applications.

Despite these advantages, there are limitations to consider. The model’s accuracy depends on the quality and representativeness of historical data. Future work could integrate satellite or UAV-based remote sensing data to capture spatial variability in soil moisture and crop conditions. Hybrid approaches, such as combining LSTM with transformer architectures or applying federated learning, could further enhance predictive performance, scalability, and privacy.

In summary, the results demonstrate that a smart irrigation system powered by LSTM and edge computing offers a highly accurate, reliable, and autonomous solution for water management. This approach optimizes irrigation, conserves water resources, and provides a practical framework for precision agriculture, particularly in areas with limited connectivity or scarce water availability. The study provides a solid foundation for further refinement and broader implementation of machine learning-based irrigation systems.

## 6. Conclusions

This paper has presented an automated irrigation system using LoRaWAN Edge-Computing irrigation Control, a solution that allows automatic watering with a CDI strategy of almond trees plantation. The system leverages weather conditions and soil moisture analysis to optimize irrigation by preventing over-watering or flooding, saving water in the process.

The solution demonstrates the feasibility of deploying autonomous, energy-efficient smart irrigation systems in agricultural areas with limited internet connectivity, leveraging edge processing to execute machine learning models locally. The platform’s flexibility allows it to integrate multiple sensors and actuators, making it suitable for diverse crops and environmental conditions.

The Arduino Edge Control is a versatile platform capable of controlling various actuators and handling a wide range of sensors and ML processing, making it suitable for deployment in harsh agricultural environments where internet network access can be an issue to rely on cloud computing. This system offers a practical, efficient solution for remote monitoring and control of irrigation systems, contributing to smarter farming practices and resource conservation.

Despite the promising results, the study is limited by the duration and geographical scope of the training data and by the absence of multi-site validation, which may affect the generalization of the model to different crops or environmental conditions. Future work will address these aspects by extending the dataset across multiple seasons and locations to strengthen the model’s robustness and transferability. In addition, the system could be further enhanced by exploring hybrid LSTM–Transformer models to capture long-term dependencies, incorporating federated learning to preserve data privacy across farms, and integrating remote-sensing information from satellites or UAVs to broaden spatial coverage.

## Figures and Tables

**Figure 1 sensors-25-07079-f001:**
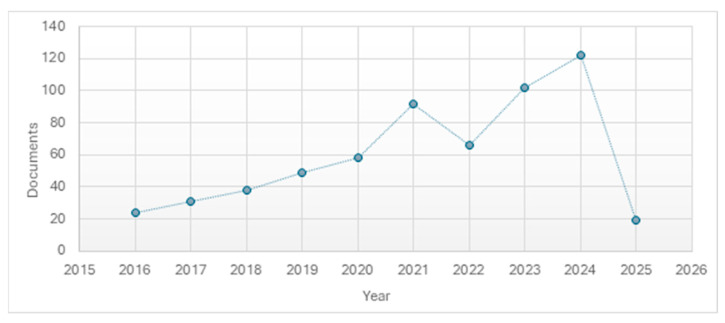
Documents on sustainable deficit irrigation by year of publication.

**Figure 2 sensors-25-07079-f002:**
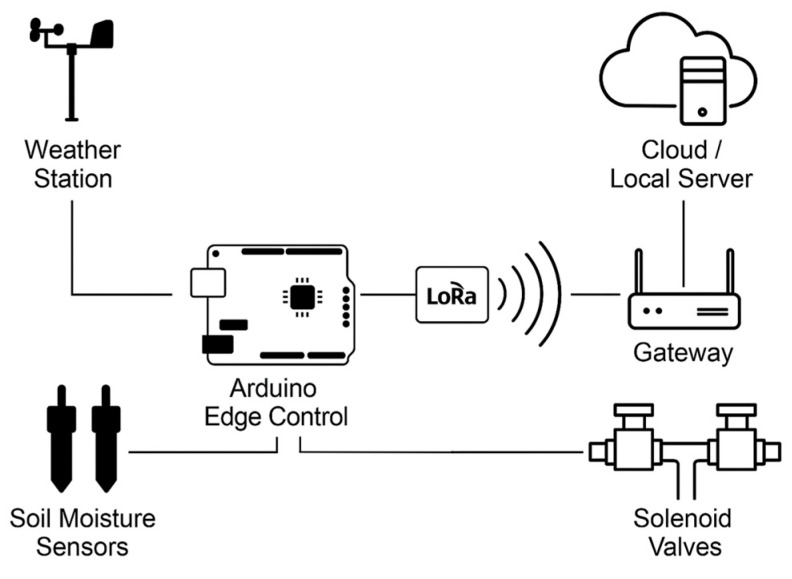
System architecture.

**Figure 3 sensors-25-07079-f003:**
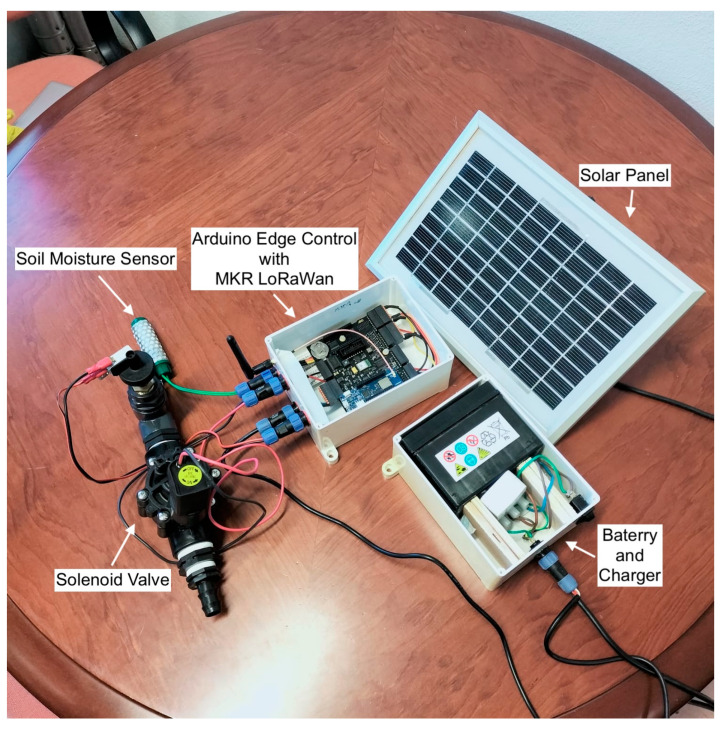
Edge node developed for the smart irrigation system.

**Figure 4 sensors-25-07079-f004:**
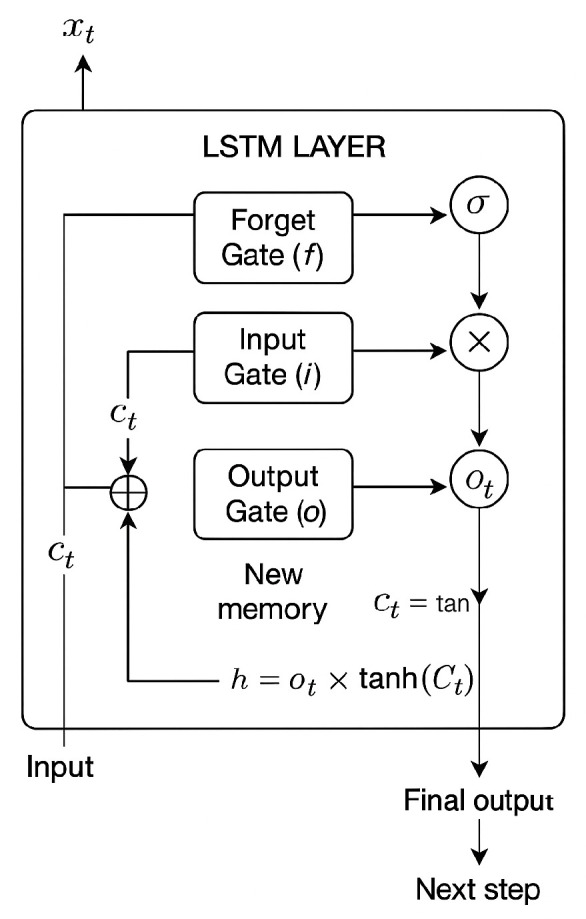
Recurrent Neural Network Structure with cell state as Input and Gate composed of a sigmoid neural net layer.

**Figure 5 sensors-25-07079-f005:**
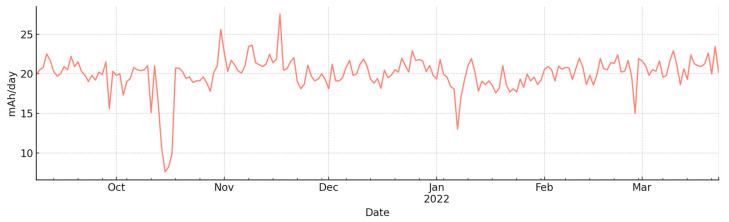
Daily energy consumption of the Arduino Edge Control node during the field campaign (from 8 September 2021 to 23 March 2022).

**Figure 6 sensors-25-07079-f006:**
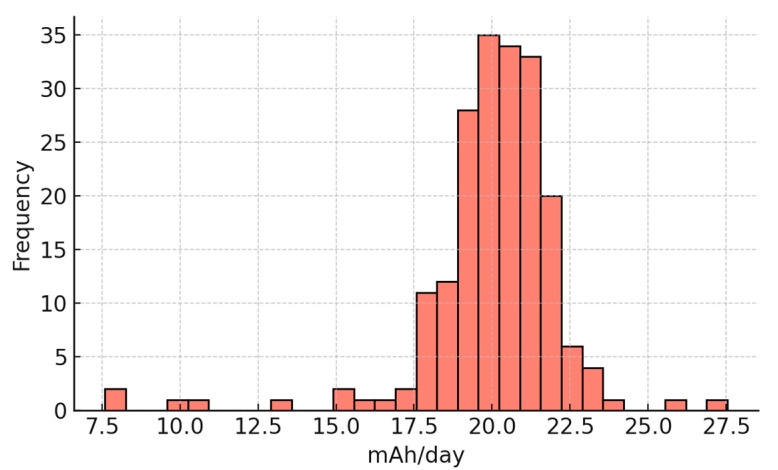
Distribution of daily energy consumption of the Arduino Edge Control node.

**Figure 7 sensors-25-07079-f007:**
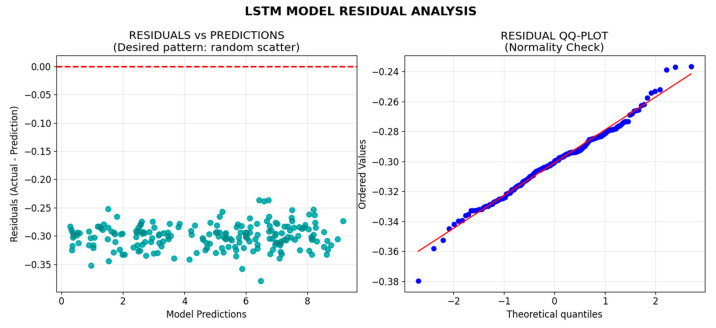
Residual analysis shows model’s assumptions are met.

**Figure 8 sensors-25-07079-f008:**
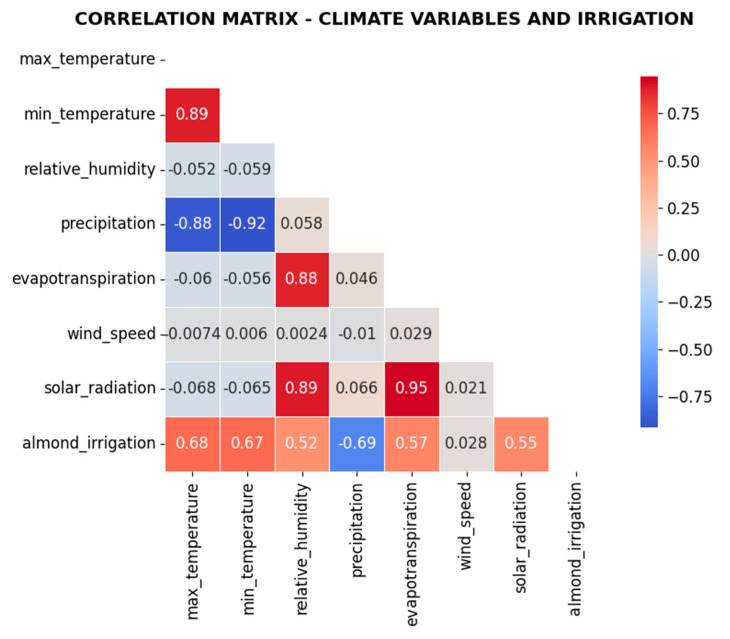
Correlation Matrix.

**Table 1 sensors-25-07079-t001:** Daily consumption metrics of the Arduino Edge Control.

Period	Total (mAh)	Avg Daily (mAh/Day)	Std Daily (mAh/Day)	Max Daily (mAh/Day)	Min Daily (mAh/Day)
8 September 2021 to 23 March 2022	3935.9	20.0	2.28	27.5	7.6

**Table 2 sensors-25-07079-t002:** Performance comparison between cloud-based and edge-based architectures.

Metric	Cloud Based	Edge Based	Improvement
Latency (ms)	450	103	77%
Response time (ms)	520	125	76%
Bandwidth usage (MB/day)	12.4	3.1	75%
Energy consumption (mAh/day)	22	20	9%

**Table 3 sensors-25-07079-t003:** Model Predictions Metrics.

LSTM Model Metrics	Value
MSE (Mean Squared Error)	0.0909
RMSE (Root Mean Squared Error)	0.3016
R^2^ (Coefficient of Determination)	0.9854

## Data Availability

The raw data supporting the conclusions of this article will be made available by the authors on request.

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
