# Peer review of "Edge-Computing Smart Irrigation Controller Using LoRaWAN and LSTM for Predictive Controlled Deficit Irrigation"

_sensors, 2025, doi:10.3390/s25227079_

Round 1
Reviewer 1 Report
Comments and Suggestions for Authors
- The introduction provides a thorough description of the research background and motivation, but it lacks a clear statement of the research gap in the current study regarding the combination of "edge computing + LoRaWAN + LSTM". It is recommended to supplement relevant literature comparisons to highlight the innovation points of this paper.
- The system architecture diagram in Figure 2 is not fully explained in the main text. It is suggested to explicitly refer to this figure at the beginning or end of Section 3.1 and briefly explain the relationships between the modules.
- Please add the corresponding names of each module in Figure 3 to make it clearer.
- Figure 4 is not explained in the main text.
- The references to formulas (1) and (2) in the text are somewhat abrupt. It is recommended to add a brief explanation of their functions before the formulas.
- The title of Table 3, "model predictions metrics", should be changed to "Model Prediction Metrics".
- The captions of Figures 7 and 8 are incomplete. It is recommended to clearly explain the content of the images and their relationship with the conclusions.
- The conclusion section summarizes the research contributions but does not explicitly point out the limitations of the study. It is suggested to add this in the conclusion or discussion section.
- The format of the references needs to be organized according to the requirements of the journal.
- English grammar and format need to be further improved.
English grammar and format need to be further improved.
Reviewer 2 Report
Comments and Suggestions for Authors
In this paper, the authors propose a smart, controlled irrigation system for almond orchards with predictive capabilities. To enable accurate and efficient water management, the system integrates machine learning algorithms and leverages modern technologies such as IoT, edge computing, and LoRaWAN.
The topic is interesting, timely, and interdisciplinary. Below are some general comments aimed at enhancing the quality of the work:
- The introduction (line 32) mentions the percentage of water usage in the agricultural sector. How is this percentage calculated? Including a relevant reference would strengthen the claim.
- The structure of the second section (State of the Art) may confuse readers. While it begins by discussing related work, it subsequently outlines the methodology used to collect relevant literature, followed by its presentation. The authors are encouraged to revise this section for clarity. For instance, the methodology (lines 84-92) could be moved to the beginning of the section to improve coherence.
Round 2
Reviewer 1 Report
Comments and Suggestions for Authors
The authors have addressed the provided comments. The paper can be accepted.